# Effect of Additives on Thermal Degradation and Crack Propagation Properties of Recycled Polyethylene Blends

**DOI:** 10.3390/polym16142060

**Published:** 2024-07-19

**Authors:** Hniya Kharmoudi, Alae Lamtai, Said Elkoun, Mathieu Robert, Carl Diez

**Affiliations:** 1Center for Innovation in Technological Eco-Design (CITE), University of Sherbrooke, Sherbrooke, QC J1K2R1, Canadamathieu.robert2@usherbrooke.ca (M.R.); 2Research Center for High Performance Polymer and Composite Systems, CREPEC, Montreal, QC H3A OC3, Canada; 3Soleno Inc. Maitrise de L’eau Pluviale, 1160 QC-133, Saint-Jean-sur-Richelieu, QC J2X 4B6, Canada; cdiez@soleno.com

**Keywords:** carbon black (CB), compatibilizer (CO), antioxidant (AO), polyethylene (PE), oxidative induction time (OIT), stress crack resistance (SCR)

## Abstract

Additives, such as antioxidants (AOs), carbon black (CB) and compatibilizers (COs), are used in recycled polymer blends for different reasons. AOs slow thermal degradation, CB gives blends a black color and protect them against ultraviolet (UV) light, and compatibilizers improve compatibility between the different phases of the mixture and consequently enhance the mechanical properties of the final blend. In this paper, the three additives were added to recycled polyethylene (PE) blends to study their effect on the final properties and to determine the best formulations that help improve the mechanical properties of recycled PE blends. Stress Crack Resistance (SCR) was accessed by performing Notched Crack Ligament Stress (NCLS) and Un-notched Crack Ligament Stress (UCLS). On the other hand, Oxidative Induction Time (OIT) was used to determine the oxidation time of the blends and the effect of each additive on this property. Based on the results of this study, it was proven that adding carbon black and antioxidants delay the thermal degradation of recycled PE blends and consequently improve the OIT. Otherwise, resistance to stress cracking is improved only by adding a compatibilizer to the reference blend.

## 1. Introduction

Plastic consumption has increased during last decade; it reached more than 460 million tonnes (Mt) in 2019 [1]. In Canada, it is growing at a rate of 8.4% each year [2]. Global plastic waste has doubled over the past years to reach 353 million tonnes (Mt) in 2019. Most Plastic Solid Waste (PSW) comes from plastics with a lifecycle under five years, such as those used in packaging, consumer goods, clothing and textiles [1].

Indeed, only 9% of plastic waste is recycled, while 19% is incinerated, and 72% ends up either in landfills or in terrestrial or aquatic environments. Plastic waste constitutes 3.4% of global greenhouse gas emissions; thus, to protect the environment, a new economical model has been introduced to valorize post-consumed plastic by promoting recycling and consequently avoid landfilling and incineration [3]. Giving plastic waste a second life is a crucial practice for reducing environmental pollution (land, water, and air), promoting sustainability and conserving natural resources.

Basically, recycling techniques can be classified into three categories: physical recycling (primary and secondary recycling), chemical recycling and energy recovery (incineration) [2,3,4]. Mechanical recycling is the most known and easiest way to recover used plastics. It is suitable for contaminated plastic to be reused to manufacture new products [5]. It includes several steps from sorting to pelletizing post-consumer plastic [4].

As part of the circular economy, several industries are recycling plastics to give them a second life. In the pipe industry, polyethylene is one of the most used materials thanks to its excellent short- and long-term mechanical properties, chemical resistance and strength. Different grades of polyethylene, such as High-Molecular-Weight Polyethylene (HMWPE), High-Density Polyethylene (HDPE), Low-Density Polyethylene (LDPE) and Linear Low-Density Polyethylene (LLDPE), are mixed to manufacture pipes [6,7].

Mechanical recycling is crucial for creating an environmentally and economically sustainable plastic economy. However, the current mechanical process faces two major challenges: the degradation of polymers and incompatibility between the different phases of polymer blending. Mixing different grades of polyethylene can result in materials with lower properties than those of the pure components [8]. Indeed, ensuring properties that meet standards is a real challenge when using recycled plastic; that is why various additives are added, such as antioxidants, compatibilizers, stabilizers, modifier agents and carbon black.

During polyolefin recycling, degradation happens, resulting in the production of various carbon-based molecules, including acids, aldehydes, ketones, alcohols, cyclic ethers and esters, as well as short-chain hydrocarbons [9]. The main polyolefins (HMWPE, HDPE, LDPE, LLDPE and PP) used in the industry have different degradation mechanisms [10]. For example, HDPE shows two mechanisms, chain scission and chain branching during extrusion, depending on process conditions [11]. It was reported in several papers that in an environment rich in oxygen, carbonyl concentration increases due to thermos-oxidative chain scission of polymer backbones via the creation of stable carbonyl end groups [11]. However, in an environment with low oxygen concentration, the shear induces chain scission and consequently blocks chain branching [12].

During continuous extrusion, polymer chains progressively decrease in length, shortening the chains and making them less susceptible to shear forces [11,13]. Crosslinking and chains scission can be overcome by reducing screw speeds and introducing additives [14]. Peroxides have been used to induce crosslinking between chains to minimize chain scission and maintain mechanical properties [15,16].

The choice of additives depends on their solubility and on the dispersion of the PE matrix [17]. Both phenols and phosphate-based antioxidants are effective in stabilizing hydrogen bonding. The use of both of them at the same time has shown important results [18].

However, the combination of hindered amine light stabilizers (HALSs) with phenolic antioxidants negatively influences the stabilization of nitroxyl radicals produced from photooxidation, reacting with phenolic groups. To overcome this problem, carbon black is used as an inorganic UV stabilizer [19]. In general, carbon black (CB) is added to the blend during pipe production to absorb UV light and to give the pipe a black color. CB is added to the blend in the form of masterbatch, containing approximately 50% of carbon black, and the rest carries polymers. In general, a concentration of 2 to 3% of CB is added to protect pipes from ultraviolet (UV) light [20].

Inorganic fillers and modifier agents can be used to enhance the mechanical properties and the thermal resistance of polymers. Inorganic fillers such as calcium carbonate improve the elastic modulus and thermal resistance, but elongation at break is reduced. Impact modifiers as elastomers enhance elongation at break and impact strength, while it reduces the elastic modulus [21,22].

Polyethylene (PE) and polypropylene (PP) are the most used commodity polymers, but their separation remains very difficult because of their similar density [23]; 8 to 10% PP is often present in PE ballots [23,24]. The immiscibility of PP and PE causes a decrease in a blend’s mechanical properties, such as elongation at break and Stress Crack Resistance (SCR) [25,26]. For this reason, compatibilizers are used in a PE blend; they are considered a bonding agent between two incompatible phases. The compatibilization mechanism depends on the components of the blend and the chemical structure of the compatibilizer. There are two principal methods to compatibilize blends: physical bounding and chemical bonding between phases [21].

This study is about the incorporation of additives in a blend composed of recycled polyethylene blends produced by melt mixing using a twin-screw extruder. Since the recycled resin will be used to produce pipes, the incorporation of additives remains necessary to improve their thermomechanical properties. As mentioned before, carbon black (CB) and antioxidants are, respectively, used to protect pipes against UV light and to enhance their Oxidative Induction Time. However, a compatibilizer is necessary to improve the miscibility between the polyethylene (PE) blends and polypropylene (PP).

The main objective of this research is to study the combined and sperate effect of carbon black, antioxidants, and compatibilizers on stress cracking resistance (SCR) by evaluating the Notched Crack Ligament Stress (NCLS), Un-notched Crack Ligament Stress (UCLS) and Oxidative Induction Time (OIT) of recycled PE.

## 2. Materials and Methods

### 2.1. Materials and Sample Preparation

The reference blend based on recycled PE was prepared in an industrial line using a twin-screw extruder. The used polyethylene resins were recycled High-Density Polyethylene (rHDPE) and recycled High-Molecular-Weight Polyethylene (rHMWPE) provided from a plastic sorting center. Table 1 shows the rate of used polyethylene (PE) grades.

Different additives were added to the reference blend: carbon black (CB), an antioxidant (AO) and a compatibilizer (CO).

The AO masterbatch contained 50% AO and 50% carrier resin; the AO was used to help enhance the processing and long-term stability of recycled plastics and to improve the durability of recycled polyolefin blends. The used CO was composed of isotactic propylene repeat units with random ethylene distribution. Some rheological properties of these additives are presented in Table 2.

The aim of this study is to test these three additives by evaluating their separate and combined effects on the thermomechanical properties of recycled High-Density Polyethylene blends, using an experimental design to test all possible additive combinations. Each additive has several levels corresponding to quantities to add to the reference blend (sample number 1: $W_{t}$ = 100%) (Table 3).

The experimental design determining the number of tests and additives combinations to be performed is illustrated in Table 4. A total of 12 tests were performed.

The first step of this study consists of extruding the reference sample without additives (number 1) via a twin-screw extruder. The second step involves adding additives to the reference sample according to the experimental design above (Table 4).

During the experiments, the RPM was fixed at 225, and the output was fixed at 1200 kg/h; the temperature in the twin-screw extruder was chosen depending on polymer blend components, as shown in the table below (Table 5).

### 2.2. Experimental Methodologies

#### 2.2.1. Notched Crack Ligament Stress (NCLS)

NCLS is a test used to control the tenacity of a material according to ASTM F2136-18 standards [27]. The main objective of the test is to access crack propagation in a controlled atmosphere. To conduct this test, a specimen is prepared from compression-molded plaques (Figure 1). This sample is then notched and submerged in a solution containing water and 10% Igepal, all at a temperature of 50 °C [27]. This test consists of placing five specimens under uniform ligament stress conditions within a constant temperature of 50 °C, attaching the weight tube to the lever arm of each specimen and recording the time-to-failure of all specimens [27].

#### 2.2.2. Un-Notched, Constant Ligament Stress Crack (UCLS)

The UCLS test is used to evaluate the response of HDPE materials containing post-consumer recycled HDPE (PCR-HDPE) to constant applied stress, since contaminants in PCR-HDPE can initiate stress cracks at an elevated temperature. The test uses the same equipment as the NCLS test [27]; the specimen is prepared from compression-molded plaque, as shown in (Figure 2). Contrary to NCLS, UCLS specimens are not notched. The test consists of five specimens attached to the load frame and placed into a water bath heated at 80 c. Weight tubes are applied to the lever arms, and timers are reset to zero to start the test. Once the test has been completed, the timer stops when a specimen fails, and the failure times are recorded. At the end of the test, if specimens fail near the grip, they are classified as a non-test if the failure happened in the reduced area [28].

#### 2.2.3. Oxidative Induction Time (OIT)

The (OIT) was measured using a TA instrument. About 10 mg of a pellet sample was used for all the blends. Samples were run through a heating cycle; they were heated in room temperature to 200 °C at a rate of 30 °C/min under nitrogen. After an isotherm time of 1 min, the gas was switched to oxygen. After induction time, an exothermic peak temperature appeared, and the time corresponding to the onset minus 12 min was considered the OIT value [29].

#### 2.2.4. Differential Scanning Calorimeter (DSC)

Thermal analysis of the blends was conducted using a differential scanning calorimeter according to ASTM-D3895-14 [29]. The melting and crystallization behaviors of polyethylene blends were measured using a TA instrument, DSC. Tests were carried out under a nitrogen flow rate of 10 mL/min. A total of 5 to 10 mg of the sample was encapsulated in an aluminum pan and heated up at 10 °C/min from the ambient temperature up to 80 °C, then held there for 1 min before heating again at 10 °C/min up to 200 °C [29].

## 3. Results and Discussion

### 3.1. Oxidative Induction Time (OIT)

The effects of adding carbon black, the antioxidant and the compatibilizer to the reference blend on the Oxidative Induction Time (OIT) are illustrated in the graphic below (Figure 3):

Adding 4% of carbon black (CB) to the reference blend increased the OIT by 180%; it increased from 10.74 to 30.2 min. This can be explained by the increase in the number of hydroxide groups that were attached to the surface of the added carbon black [30], while adding the same amount of CB to the blend containing 3% of the compatibilizer (CO) increased the OIT by 73%; it increased from 12.55 to 21.77 min, as well as for the blend with 0.5% of AO.

OIT enhancement was more interesting when it came to add CB to blends containing only an antioxidant as an additive. On the other hand, the improvement was less significant when CB was added to blends containing AO and a compatibilizer.

For carbon black, the effect on the OIT was more significant than adding only a compatibilizer or antioxidant. This result can be explained by the chemical composition of each additive. The carbon black was composed of 50% CB and 50% of a carrier (PE), while the antioxidant was composed of 2.5% antioxidant and the carrier (97.5).

### 3.2. Notched Crack Ligament Stress (NCLS)

The effects of adding carbon black, the antioxidant and the compatibilizer to the reference blend are illustrated in the graphic below.

The effect of additives on the NCLS of PE blends is illustrated in Figure 4. Adding carbon black (CB) to the reference blend decreased NCLS by 8%, while adding 3% of compatibilizer (CO) to the reference blend increased NCLS by 16.7% and adding carbon black (CB) to this blend decreased NCLS by 22.25%. Adding the compatibilizer (CO) increased the internal interfaces between phases of the blends [31,32].

The NCLS of blends was enhanced compared to the reference blend while adding 3% of the compatibilizer (CO), 0.5% of the antioxidant (AO) and a mix of the two additives by 16.7%, 3.4% and 8%, respectively, while adding CB decreased the NCLS no matter which additives were added. The decrease in NCLS while adding CB can be related to the heterogeneous dispersion of this additive in the matrix [33,34,35].

### 3.3. Un-Notched Crack Ligament Stress (UCLS)

The effects of adding carbon black, the antioxidant and the compatibilizer to the reference blend are illustrated in the graphic below.

The effect of additives on the UCLS of the reference blend is shown in Figure 5. Adding a CB to the reference blend decreased the UCLS by 37%, while adding 3% of the compatibilizer to the reference blend increased UCLS by 15.35%. A decrease of 39.14% was observed by adding 4% CB to the blend containing 3% of the compatibilizer. All blends showed a decrease in UCLS compared to the reference blend. Adding 0.5% of an AO to the reference blend decreased UCLS by 34.42% and adding 4% of CO to the mixture containing 0.5% of an AO reduced UCLS by 44.5% compared to the reference mixture. Adding a mix of 0.5% of an AO and 3% of CO slightly increased the UCLS compared to blends with only a compatibilizer or antioxidant, but values remained lower than the UCLS values of the reference blend. The addition of 3% of CB to the mix of 0.5% of an AO and 3% of a CO decreased the UCLS more, by 32.6% compared to the reference blend. The point to be made is that the addition of 4% of CB decreased the UCLS of the blends more, compared to blends with only compatibilizers or antioxidants as additives.

The decrease in UCLS while adding CB is also related to the heterogeneous dispersion of this additive in the matrix and potential contaminants present in the blend since the PE used was recycled [35].

### 3.4. Differential Scanning Calorimetry (DSC)

Differential Scanning Calorimetry was carried out to study the thermal behavior of the blends. The thermal results (Table 6) obtained by DSC are summarized in the following question, which presents the degree of crystallinity of the blends.
$$\chi_{c}=\frac{{\Delta H}_{f}}{{\Delta H}_{f}^{0}}$$
where ${\Delta H}_{f}$ is the enthalpy of fusion of the sample measured during th test (in J/g), and the ${\Delta H}_{f}^{0}$ presents the enthalpy of fusion of a 100% crystalline polyethylene estimated in the literature at 293 J/g.

The degree of crystallinity of all blends are within the following range [39–60%]; this proved the semi-crystallinity of the mixtures. The minimum value of 39% returns to the blend compatibilized with 3% of CB, and the maximum value of 58% returns to the blends composed of 0.7% AO, 4% carbon black and 3% CB.

Adding 0.5 and 0.7 of the antioxidant increased the degree of crystallinity, respectively, by 3% and 7%, compared to the reference. Mixing the antioxidant with 4% carbon black increased the degree of crystallinity by 5% compared to the reference.

These results show that the addition of a compatibilizer decreased the degree of crystallinity, while the addition of carbon black and antioxidants increased the degree of crystallinity.

## 4. Conclusions

In this work, polymer blends based on recycled HDPE and HMW were prepared via a twin-screw extruder using three different additives:-A compatibilizer;-Antioxidants;-Carbon black.

The thermal degradation and crack propagation properties of recycled polyethylene blends were assessed, and the following conclusions were obtained:For thermal degradation, the mixing of carbon black and antioxidants, or antioxidants and compatibilizers, multiplied the Oxidative Induction Time (OIT) by a factor of 7 compared to the reference blend without any additives.The addition of carbon black negatively impacts stress cracking, while adding an antioxidant with a compatibilizer improved the time before failure in the NCLS.No matter which additive was used, UCLS decreased compared to the reference blend.This study showed that the addition of a compatibilizer decreases the degree of crystallinity; however, the addition of carbon black and antioxidants increases the degree of crystallinity.

## Figures and Tables

**Figure 1 polymers-16-02060-f001:**
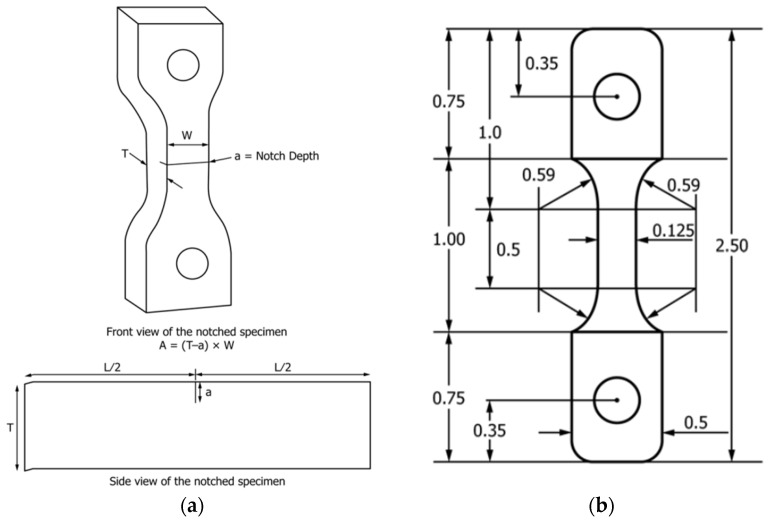
(**a**). Notching position. (**b**). Specimen geometry [27].

**Figure 2 polymers-16-02060-f002:**
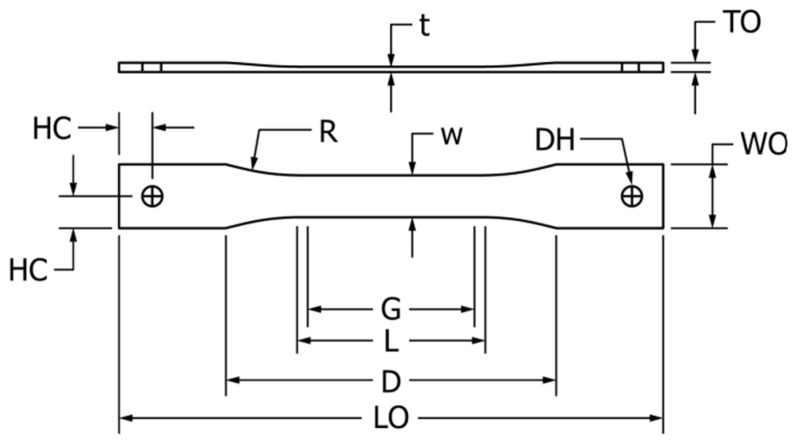
UCLS test specimen [28].

**Figure 3 polymers-16-02060-f003:**
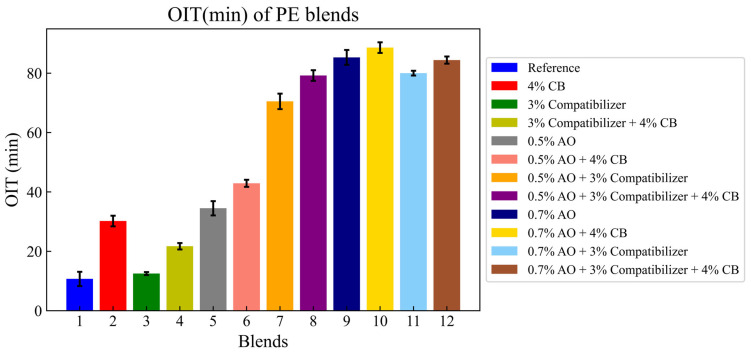
OIT (min) of PE blends.

**Figure 4 polymers-16-02060-f004:**
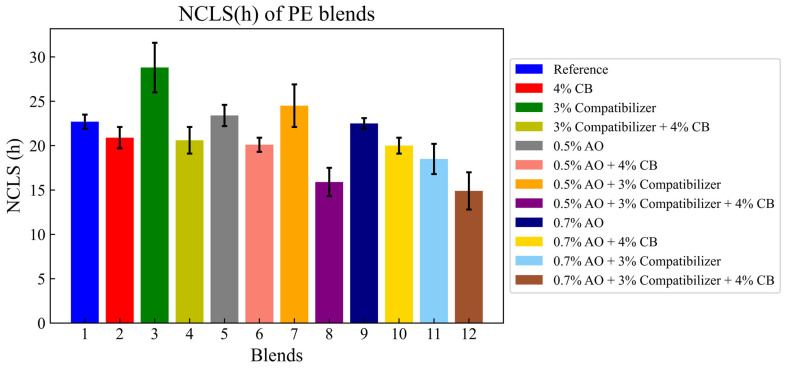
NCLS of PE blends.

**Figure 5 polymers-16-02060-f005:**
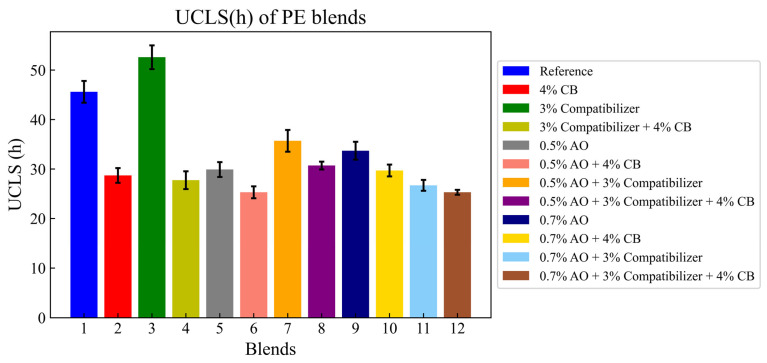
UCLS of PE blends.

**Table 1 polymers-16-02060-t001:** Reference blend.

Composition	Rate (%)
rHMWPE	65
rHDPE	35

**Table 2 polymers-16-02060-t002:** Rheological properties of used materials.

Polymer Type	MFI (g/10 min) 190 °C/2.16 kg	MFI (g/10 min) 190 °C/21.6 kg	Density (g/cm^3^)
HDPE	0.5	3–5	0.951
HMW	0.1	-	0.948
Antioxidant	1.4	-	0.862
Compatibilizer	-	-	0.47
Carbon black	-	6–13	-

**Table 3 polymers-16-02060-t003:** Levels of the used additives.

Additives	Levels	Quantities (wt %)
Antioxidant	3	0 0.5 0.7
Compatibilizer	2	0 3
Carbon black	2	0 4

**Table 4 polymers-16-02060-t004:** Experimental design.

Number of Blends	Antioxidant (wt %)	Compatibilizer (wt %)	Carbon Black (wt %)
1	0	0	0
2	0	0	4
3	0	3	0
4	0	3	4
5	0.5	0	0
6	0.5	0	4
7	0.5	3	0
8	0.5	3	4
9	0.7	0	0
10	0.7	0	4
11	0.7	3	0
12	0.7	3	4

**Table 5 polymers-16-02060-t005:** Extruder temperatures.

T (°C)	200	210	215	225	235	220
Zone	1	2	3	4	5–7	7–12

**Table 6 polymers-16-02060-t006:** Degree of crystallinity of blends.

Sample Number	Degree of Crystallinity (%)
1 (Reference)	55
2	57
3	39
4	48
5	57
6	58
7	55
8	52
9	59
10	58
11	57
12	60

## Data Availability

The original contributions presented in the study are included in the article, further inquiries can be directed to the corresponding author.

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
