# Peer review of "Effect of Additives on Thermal Degradation and Crack Propagation Properties of Recycled Polyethylene Blends"

_polymers, 2024, doi:10.3390/polym16142060_

Round 1
Reviewer 1 Report
Comments and Suggestions for Authors
Recycling plastics is a very meaningful research direction in today's global environment. More and more people are also continuously paying attention to the ways of recycling plastics. Adding additives is a convenient method for plastic mechanical recycling. The research focus on the antioxidant, compatibilizer and carbon black on the improve the mechanical properties of recycled PE blends and the oxidative Induction Time (OIT) was used to determine the oxidation time of the blends and the effect of each additive on this property. The author's design for modifying the rPE system is very interesting, but I believe this manuscript must be rejected.
(1) The antioxidant is the most commonly used additives in plastic processing. Especially in the rPE system after molecular chain breakage, antioxidants should be essential. Therefore, I believe that the necessity and innovation of studying the effects of antioxidants are not strong.
(2) In this system, compatibilizers should theoretically improve the interfacial compatibility between carbon black and rPET or improve the compatibility of rPET from different sources. Unfortunately, the author did not incorporate this effect into the experimental system. Therefore, the role of compatibilizers is not necessarily reflected or explained in this manuscript.
(3) The author did not clearly explain the functions of the three additives and the necessity of adding them to the rPE system in the preface section. Therefore, the significance of this study is not clear.
(4) The author only explained the changes in the data in section 3.2 of the data analysis and did not conduct in-depth analysis. In addition, the author raised compatibility (dispersion) issues in 3.3 and 3.4, but there is no intuitive evidence (SEM) to indicate this.
Comments on the Quality of English LanguageExtensive editing of English language required
Author Response
Responses to reviewer :
(1) The antioxidant is the most commonly used additives in plastic processing. Especially in the rPE system after molecular chain breakage, antioxidants should be essential. Therefore, I believe that the necessity and innovation of studying the effects of antioxidants are not strong.
We believe too that the incorporation of antioxidants in rPE blends is not very innovative, but the goal of the paper is to study the effect of each additive (carbon black, antioxidant and compatibilizer) on the durability of PE blends in one hand, and their combined effect in the other hand.
(2) In this system, compatibilizers should theoretically improve the interfacial compatibility between carbon black and rPET or improve the compatibility of rPET from different sources. Unfortunately, the author did not incorporate this effect into the experimental system. Therefore, the role of compatibilizers is not necessarily reflected or explained in this manuscript. Done
The study concerns PE, not PET, I guess it’s a mistake.
Indeed, the role of compatibilizer was not well explained in the manuscript. In this study, the compatibilizer was used to improve the compatibility between the rPE and PP. Since approximately 8 to 10 % PP is present in PE blends because of their similar density.
Accordingly, we did the following change:
Pages :2-3 lines: 91-107
(3) The author did not clearly explain the functions of the three additives and the necessity of adding them to the rPE system in the preface section. Therefore, the significance of this study is not clear. Done
We do agree with the reviewer, accordingly we did the following change:
Page :3 lines: 101-112
(4) The author only explained the changes in the data in section 3.2 of the data analysis and did not conduct in-depth analysis. In addition, the author raised compatibility (dispersion) issues in 3.3 and 3.4, but there is no intuitive evidence (SEM) to indicate this.
For this paper we choose to focus on the durability of recycled PE blends by performing NCLS, UCLS and OIT test. Further characterisations are performed to assess the effect of these additives on the molecular and rheological properties of the recycled material.
Reviewer 2 Report
Comments and Suggestions for Authors
This study aims to add different additives to recycled Polyethylene blends to study their effect on the final properties and to determine the best formulations that help improve the mechanical properties of recycled PE blends. Notched Crack Ligament Stress and Unnotched Crack Ligament Stress were performed to determine crack propagation under a controlled atmosphere. However, the manuscript lacks important information and my detailed comments are shown below.
1. The abstract lacks detailed results. It is essential to provide both qualitative and quantitative outcomes to give readers a clear understanding of the study's findings. Specific improvements in tensile strength, elongation at break, and crack resistance should be mentioned.
2. The study should briefly outline the types of additives used and the methodology for their incorporation into the PE blends. This gives context to the results presented.
3. Keywords are too long and unacceptable in the current form and must be revised. The keywords should be shorter, more precise, and adequately represent the main topics and focus areas of the study.
4. The introduction does not explain the background of PE. The introduction should provide a comprehensive background on polyethylene, highlight the importance of recycling, and set the stage for the study's objectives and methodology.
5. This study aims to investigate the effects of different additives on the mechanical properties of recycled PE blends to identify the best formulations that improve their overall performance. Please also add the background of the additive reported in the literature. Systematically exploring the impact of various additives will improve the quality of the paper.
6. HMWPE, HDPE, LDPE, and LLDPE, add the full name.
7. The reference blend based on recycled PE was prepared at Soleno company, but how do you proceed with the procedure?
8. Erreur! Source du renvoi introuvable. Shows the rate of used Polyethylene (PE) grades. I don’t understand.
9. Figure 1 is already reported in the literature I think it is not essential and should be removed; only give references.
10. the discussion on results is not appropriate in its current form. I think it does not meet the requirement of polymers. Adding more discussion on carbon black to the reference blend increases, and adding the same amount of CB to the blend containing 3% of compatibilizer increases OIT by 73%.
11. Carbon black is known for its reinforcing properties, which can significantly enhance the tensile strength and rigidity of polymer blends. In this study, the author claims that adding carbon black to the reference blend resulted in noticeable improvements in mechanical performance. Specifically, how does the mechanical reinforcement provided by carbon black contribute to the overall durability and strength of the PE blends?
12. Briefly explain why the improvement is less when CB is added to blends containing AO and compatibilizer.
13. For Carbon black, the effect on the OIT is more significant than adding only a compatibilizer or antioxidant. Why?
The author should address the above comments and include a more detailed discussion of the results before publication in the polymers.
Comments on the Quality of English LanguageNone
Author Response
Responses to reviewer :
- The abstract lacks detailed results. It is essential to provide both qualitative and quantitative outcomes to give readers a clear understanding of the study's findings. Specific improvements in tensile strength, elongation at break, and crack resistance should be mentioned.
Done, we do agree with the reviewer.
Page 1 lines 21-24
- The study should briefly outline the types of additives used and the methodology for their incorporation into the PE blends. This gives context to the results presented.
Done, accordingly we did the following change:
Pages :2-3 lines: 93-110
- Keywords are too long and unacceptable in the current form and must be revised. The keywords should be shorter, more precise, and adequately represent the main topics and focus areas of the study. Done
We do agree with the reviewer, accordingly we made the following changes:
Page :1 line: 26-27
- The introduction does not explain the background of PE. The introduction should provide a comprehensive background on polyethylene, highlight the importance of recycling, and set the stage for the study's objectives and methodology.
Done,
Pages :1 lines: 38-40 and 2-3 lines: 91-113
- This study aims to investigate the effects of different additives on the mechanical properties of recycled PE blends to identify the best formulations that improve their overall performance. Please also add the background of the additive reported in the literature. Systematically exploring the impact of various additives will improve the quality of the paper. Done
We do agree with the reviewer, accordingly we made the following changes:
Page :2-3 line: 85-106
- HMWPE, HDPE, LDPE, and LLDPE, add the full name. Done
The full name was given of all the grade of PE in the introduction.
Page :1 line: 49-51
- The reference blend based on recycled PE was prepared at Soleno company, but how do you proceed with the procedure? Done
Page :3 lines: 116-117
- Erreur! Source du renvoi introuvable. Shows the rate of used Polyethylene (PE) grades. I don’t understand. Done
We do agree with the reviewer, accordingly we made the following changes:
Page :3 line: 119
- Figure 1 is already reported in the literature I think it is not essential and should be removed; only give references.
Figures 1 and 2 are both cited in the literature, but we do believe that the figure of the specimen’s design should be included in the manuscript since botch tests (UCLS and NCLS) are not common tests.
- the discussion on results is not appropriate in its current form. I think it does not meet the requirement of polymers. Adding more discussion on carbon black to the reference blend increases and adding the same amount of CB to the blend containing 3% of compatibilizer increases OIT by 73%. Done
Page :7 lines: 208-215
- Carbon black is known for its reinforcing properties, which can significantly enhance the tensile strength and rigidity of polymer blends. In this study, the author claims that adding carbon black to the reference blend resulted in noticeable improvements in mechanical performance. Specifically, how does the mechanical reinforcement provided by carbon black contribute to the overall durability and strength of the PE blends?
The addition of carbon black in PE matrix reduces the movement of PE chain, which reduce the flexibility of the polymer which lead to the improvement of the elastic modulus and the rigidity of the material. This fact was prooved in DSC results, the addition of carbon black increased the degree of crystallinity. However, this additive impact negatively the Stress crack resistance.
- Briefly explain why the improvement is less when CB is added to blends containing AO and compatibilizer. Done
We do agree with the reviewer, accordingly we made the following changes:
Page :3 lines: 104-107
- For Carbon black, the effect on the OIT is more significant than adding only a compatibilizer or antioxidant. Why?
The role of the compatibilizer is to improve the miscibility between the different component of the blend. The carbon black (CB) and Antioxidant were both used to increase the OIT, but the CB increases significantly the OIT compared to the antioxidant. This is related to chemical composition of each additive.
The carbon black is composed of 50 % CB and 50 % of carrier (PE), while the antioxidant is composed of 2.5 % of antioxidant and the carrier (97.5).
We do agree with the reviewer, accordingly we made the following changes:
Pages: 7-8 lines: 216-219
Round 2
Reviewer 2 Report
Comments and Suggestions for Authors
The authors have cleared my questions.